# Fault-Tolerant Event-Triggrred Control for Multiple UAVs with Predefined Tracking Performance

**Ziyuan Ma, Huajun Gong *** and **Xinhua Wang**

The College of Automation Engineering, Nanjing University of Aeronautics and Astronautics, Nanjing 211100, China; maziyuan@nuaa.edu.cn (Z.M.); xhwang@nuaa.edu.cn (X.W.)

* Correspondence: ghj301@nuaa.edu.cn

**Abstract:** This paper proposes an event-triggered fault-tolerant time-varying formation control method dedicated to multiple unmanned aerial vehicles (UAVs). We meticulously design a formation-tracking controller with a predefined tracking performance to accommodate the presence of actuator faults and external disturbances. Firstly, the formation-tracking controller acquires the desired heading using the line-of-sight algorithm. Secondly, in the presence of actuator faults and external disturbances, we introduce the radial basis function neural network (RBFNN) and adaptive law tracking control to effectively compensate for their effects. Additionally, we design adaptive tracking controllers and event-triggering conditions to increase the computational frequency. The predefined tracking performance, implemented via a Lyapunov function, ensures the convergence of the tracking error over time. Finally, we conduct a thorough analysis of the system's stability, successfully eliminating the possibility of Zeno behavior. The simulation results thoroughly validate the effectiveness of the theoretical analysis.

**Keywords:** multiple UAVs; time-varying formation control; fault-tolerant control; event-triggered control; external disturbances; predefined tracking performance





## 1. Introduction

With the increasing complexity of aviation missions and the continuous development of aircraft control technology, network communication technology, and artificial intelligence, unmanned aerial vehicles (UAVs) have achieved an overall improvement in system efficiency via the complementary capabilities and coordinated behavior among individual UAV units. In various complex tasks and scenarios such as collaborative target observation, area monitoring, cooperative analysis, and decision making, as well as collaborative formation flying and operations, UAV clusters demonstrate significant advantages. However, as the number of cluster units increases and functional relationships become more coupled, the complexity of navigation and communication grows exponentially. The formation and maintenance of formations in swarm flight missions also face greater challenges. If the UAV swarm cannot autonomously respond to unexpected failures and external disturbances under extreme weather conditions, it becomes difficult to fully leverage the advantages of collaborative operations. Therefore, higher requirements are placed on UAV swarm control, anti-interference capabilities, and fault-tolerant technologies. Formation control is important for advancing the development of multi-agent systems as it enables the multi-agent swarm to operate smoothly. For this purpose, different formation control methods have been proposed like the cooperative control method [1], adaptive control method [2,3], finite-time control method [4], robust formation method [5,6], and so on. Formation control is the crucial aspect of cooperative control in UAVs, where multiple UAVs are grouped together to perform specific tasks like cooperatively observing a specific area [7], rescue operations [8], and monitoring [9]. In [10], a model predictive control method was proposed for the formation control of multiple UAVs. In [11], a novel swarm

intelligence approach was introduced for coordinating the control of multiple drones. In [7], a multiple-UAV collaborative search and attack method was proposed. In [12], the problem of perturbation suppression for multiple high-order agent formations with directed graphs was studied. In addition, there are some classic formation control methods, such as the leader–follower formation control [13,14]. Researchers associated with control and system domain have primarily focused on proposing control algorithms for different types of UAV's applications like multiple fixed-wing UAVs [15]. In case of actual flight, fixed-wing UAVs are prone to parametric uncertainties and perturbations in the aerodynamic environment [16].

Currently, different methods have been proposed to deal with external interference and disturbances like back-stepping control [17,18], the barrier Lyapunov function (BLF) [19], neural network control [20,21], sliding mode control [22], and active disturbance rejection control [23]. In back-stepping control, the "explosion of complexity" problem can be caused by the repeated differentiation of the virtual control [24], which can make it difficult to find the optimal solution of the problem, especially in cases of uncertainties and disturbances. BLF-based adaptive finite-time control was proposed in [25], which guaranteed the robust response in the presence of uncertainties. Anti-disturbance exact flight control based on the BLF-based back-stepping technique with the mash-up of a high-order sliding mode observer was proposed for ultra-low-altitude airdrop [26], but it did not consider the actuator fault. In the field of neural network control, reference [27] proposes an innovative control strategy, namely observer-based adaptive fuzzy finite-time attitude control. This approach utilizes an adaptive neural network observer to estimate angular velocity information with finite-time characteristics. By introducing an adaptive fuzzy logic system (FLS), it successfully compensates for lumped disturbance and achieves online adjustment of control gains. Ref. [28] presents a low-computation learning-based antisaturation fixed-time attitude-tracking control method. This approach constructs a fixed-time state observer to accurately estimate the system state. Through the adoption of adaptive neural network technology, it effectively overcomes the negative impact of external disturbances and uncertain parameters on the system. It is noteworthy that the adaptive mechanism in the design achieves a reduction in the computational burden and structural simplification by online adjusting a virtual parameter rather than the weight vector of the neural network. Similarly, a continuous-time non-linear system was proposed [29] which possessed uncertainties and disturbances, but the drawback was that it did not take fault tolerance into account. Moreover, a multivariable adaptive control-based consensus flight control method with parametric uncertainties and unknown external disturbances was proposed for formation UAVs [30]. These controls have downsides in term that although they take the uncertainties or disturbances into account [25], they did not consider the actuator faults in the formation control [24,26]. These methods are using back-stepping control [24], BLF-based adaptive finite-time control [25], and BLF-based back-stepping control [26] to deal with uncertainties and disturbances.

In formation control, actuator failure refers to the situation where one or more agents in a formation experience a malfunction or failure in their actuator, which is a component responsible for generating and controlling the motion of the agent. Actuator failure can have a significant impact on the performance and stability of a formation control system [31]. In particular, it can cause the affected agent to deviate from the desired formation and potentially collide with other agents or obstacles in the environment. Moreover, the failure can propagate to other agents in the formation and lead to a cascading effect, destabilizing the entire formation. A BLF-based learning control approach in [32] and a BLF-based adaptive full-state constrained control law in [33] was proposed, considering the actuator fault. Different machine learning techniques like the neural network is applied to deal with the uncertainty and non-linearity of the system [34,35]. An event-triggered extended state observer (ESO) was designed, which only triggered output signals and was independent of the system states. An event-triggered active disturbance rejection control (ADRC) strategy was introduced in [36] which did not bound the latest state, disturbance

estimates, and control signals to rely on any sensor-controlled network transmission until the violation of the discrete triggered condition. Moreover, for high-order linear multi-agent systems, fault-tolerant time-varying formation control problems were investigated in the presence of actuator faults [37]. Limitation of the aforementioned control methods was that although they considered the actuator faults, they lack the ability to take the uncertainties into account [37]. If they did so, then they did not bound the disturbance estimates [36].

In recent advances, tasks are becoming more and more complex. In order to ensure the flight safety of the formation, researchers must consider both external interference and actuator failure. To remove the problems in prior methods, a composite decentralized fractional-order back-stepping adaptive neural fault-tolerant control (FTC) method, which was the amalgamation of neural networks, disturbance observers, fractional calculus, and high-order sliding-mode differentiators, was proposed for the attitude synchronization tracking of multiple UAVs. Its objective was to address the actuator fault and wind effect problem [38]. Using a distributed sliding-mode estimator, dynamic surface control architecture, neural networks, and disturbance observers, a distributed adaptive fault-tolerant control scheme was proposed in the presence of actuator faults and wake vortices [22]. Similarly, based on the local information of neighboring UAVs, a fault-tolerant cooperative controller was designed considering actuator faults, input saturation, and external disturbances [39]. In [40], a fault-tolerant time-varying elliptical formation control scheme was proposed by using a fractional-order sliding mode control strategy. The aim was to monitor the elliptical spread of forest fire using multiple UAVs. Moreover, the sliding-mode disturbance observer comprising reference systems and sliding-mode differentiators was proposed to estimate the lumped disturbances caused by the external disturbances and actuator faults. The problem of actuator faults, input saturation, and the wake vortex effect was investigated for the safe control of trailing UAV and was estimated using disturbance observers in [41], and then using estimated disturbance, back-stepping control laws were developed for longitudinal and lateral-directional dynamics. One of the key aspects of this control was that it considered the external wake vortex, disturbances, internal actuator faults, and input saturation at the same time. The above control methods considered the actuator failure, external interference, and disturbance simultaneously and used the disturbance observer [22,38,41], fault-tolerant cooperative control [39], and fault-tolerant time-varying elliptical control [40] for that purpose. To ensure the flight safety in a complex task environment, there is still room for more and more development.

This paper proposes a fault-tolerant time-varying formation control for multiple UAVs with a predefined tracking performance, taking into consideration the limitations of the existing literature. The guidance law for formation tracking is designed, and the performance of the tracking error is pre-determined using a Lyapunov function. An adaptive tracking controller and speed controller are designed, and the stability of the system is analyzed. The main contributions of this work, compared with the existing methodologies, are as follows:

- The guidance rate of formation control is designed using a line-of-sight (LOS) guidance algorithm, and the performance of the formation-tracking error is realized using a Lyapunov function. This approach is compared with the method proposed in [36], and it provides better performance by bounding the convergence error.
- A sampling adaptive tracking controller is proposed for the velocity and yaw angle loop, in combination with the radial basis function neural network (RBFNN). This approach performs better in case of actuator failure and external disturbances, with a reduction in communication and actuation consumption compared to the work presented in [38].
- A novelty sampling mode with an event-triggering mechanism is designed, which utilizes only the input information to implement the sampling. This approach avoids the need for dedicated monitoring devices, as seen in the triggering mode presented in [29], and reduces the communication burden of the actuation.

The remaining sections of the paper are organized as follows. Section 2 describes the modeling and provides some prerequisite knowledge. Section 3 proposes the guidance law design. Section 4 presents comparative simulation results. Finally, Section 5 concludes our work.

## 2. Problem Statement

### 2.1. Dynamic Modeling of the Multiple Non-Linear UAVs

The twelve differential equations of the UAV six-degree-of-freedom model are as follows [42]:

$$
\begin{cases}
m\dot{V} = T\cos\alpha - D - mg(\cos\alpha\sin\theta - \sin\alpha\cos\beta) \\
mV\dot{\beta} = Y - mV(-p\sin\alpha + r\cos\alpha) \\
mV\dot{\alpha} = -T\sin\alpha - L + mVq + mg(\sin\alpha\sin\theta + \cos\alpha\cos\theta) \\
\dot{\phi} = p + (r\cos\phi + q\sin\phi)\tan\theta \\
\dot{\theta} = q\cos\phi - r\sin\phi \\
\dot{\psi} = \frac{1}{\cos\theta}(r\cos\phi + q\sin\phi) \\
\dot{p} = (c_1 r + c_2 p)q + c_3\bar{L} + c_4 N \\
\dot{q} = c_5 pr - c_6(p^2 - r^2) + c_7 M \\
\dot{r} = (c_8 p - c_2 r)q + c_4\bar{L} + c_9 N \\
\dot{x}_g = u\cos\theta\cos\psi + v(\sin\phi\sin\theta\cos\psi - \cos\phi\sin\psi) \\
\quad\quad + w(\sin\phi\sin\psi + \cos\phi\sin\theta\cos\psi) \\
\dot{y}_g = u\cos\theta\sin\psi + v(\sin\phi\sin\theta\sin\psi + \cos\phi\cos\psi) \\
\quad\quad + w(-\sin\phi\cos\psi + \cos\phi\sin\theta\sin\psi) \\
\dot{h} = u\sin\theta - v\sin\phi\cos\theta - w\cos\phi\cos\theta
\end{cases}
\tag{1}
$$

where

$$
\begin{bmatrix} u \\ v \\ w \end{bmatrix} = S_{\alpha\beta}^T \begin{bmatrix} V \\ 0 \\ 0 \end{bmatrix}_{\text{wind}} = \begin{bmatrix} V\cos\alpha\cos\beta \\ V\sin\beta \\ V\sin\alpha\cos\beta \end{bmatrix}
\tag{2}
$$

$$
c_1 = \frac{(I_y - I_z)I_z - I_{xz}^2}{\Sigma}, c_2 = \frac{(I_x - I_y + I_z)I_{xz}}{\Sigma}, c_3 = \frac{I_z}{\Sigma}, c_4 = \frac{I_{xz}}{\Sigma}, c_5 = \frac{I_z - I_x}{I_y},
$$

$$
c_6 = \frac{I_{xz}}{I_y}, c_7 = \frac{1}{I_y}, c_8 = \frac{I_x(I_x - I_y) + I_{xz}^2}{\Sigma}, c_9 = \frac{I_x}{\Sigma}, \Sigma = I_x I_z - I_{xz}^2
\tag{3}
$$

$u, v, w$ respectively represent the three directions of the UAV in the body coordinate system, $I_x, I_y, I_z$ are respectively the moment of inertia of UAV around three body axes, and $I_{xz}$ is the product of inertia.

Thus, for multiple UAVs, the non-linear mathematical model is described by the following generalized set of non-linear differential equations:

$$
\begin{cases}
\dot{x} = f(t, x, u) \\
y = g(t, x, u)
\end{cases}
\tag{4}
$$

In the equation, $t$ represents time, $x$ represents the system state, $u$ represents the control input, and $y$ represents the output. The state variables' airspeed, angle of attack, sideslip angle, roll angle, pitch angle, heading angle, roll rate, pitch rate, yaw rate, vertical displacement, lateral displacement, and altitude are represented as $x = [V, \alpha, \beta, \phi, \theta, \psi, p, q, r, x_g, y_g, h]$, and the control variables $u = [T, D, Y, L, M, N]$ represent thrust, drag, side force, rolling moment, pitching moment, and yawing moment, respectively. These control variables are functions of aerodynamic coefficients.

In this article, if the formation control of a UAV is only concerned with positions and velocities, then an inner/outer loop structure can be used to implement the formation control [43,44]. This is because the trajectory dynamics of the UAV has much larger time constants than the attitude dynamics. In this configuration, the outer loop is used to drive

the UAV towards the desired position with the desired velocity, while the inner loop is used to track the attitude. The focus of this brief is on designing the outer loop.

At the formation control level, a UAV can be approximated as a point–mass system, and its dynamics can be simplified and described by the following equations referring to [45]:

$$
\begin{cases}
\dot{x}_i = u_i \cos \psi_i \\
\dot{y}_i = u_i \sin \psi_i \\
\dot{\psi}_i = r_i \\
\dot{u}_i = T_i + \vartheta_{T,i} \\
\dot{r}_i = \tau_i + \vartheta_{\tau,i}
\end{cases}
\tag{5}
$$

for $i = 1, \ldots, n$, where $\boldsymbol{p_i} = [x_i, y_i]^{\mathrm{T}}$ is the UAV position, $u_i$ and $r_i$ are the airspeed and yaw angle rate of the UAV, respectively, $\psi_i$ is the yaw angle, $T_i$ is the thrust force, $\tau_i$ is the yawing moment, and nominal disturbances $\boldsymbol{\vartheta}_i = [\vartheta_{T,i}, \vartheta_{\tau,i}]^{\mathrm{T}} = [\Delta T_i + d_{u,i}, \Delta \tau_i + d_{r,i}]^{\mathrm{T}}$. $\Delta T_i$ and $\Delta \tau_i$ are, separately, the thrust force error and the yawing moment error caused by the actuator fault. $d_{u,i}$ and $d_{r,i}$ are external disturbances. As a general consideration, for all time $k$, the nominal disturbance $\boldsymbol{\vartheta}_i$ is bounded by $|\boldsymbol{\vartheta}_i(k)| \leq \vartheta_0$, where $\vartheta_0$ is a positive constant.

Some useful lemmas are given:

**Lemma 1 ([46]).** *For any given continuous function $f(x)$ with $f(0) = 0$ defined on the compact set $\Omega_x$, through the continuous function separation and RBFNNs approximation techniques, $f(x)$ can be ultimately modeled as*

$$
f(x) = AS(x) + \varepsilon(x), \forall x \in \Omega_x
$$

*where $\varepsilon(x)$ is the approximation error satisfying $|\varepsilon(x)| \leq \bar{\varepsilon}$. $S(x) = (S_1(x), S_2(x), \ldots, S_l(x))^T$ denotes the Gaussian basis function vector with $S_j(x) = \exp\left(-\frac{(x-e)^T(x-e)}{2\sigma^2}\right), j = 1, 2, \ldots, l,$ where $\varrho$ and $\sigma$ are the center and width of Gaussian basis function $S_j(x)$, respectively. $A$ is the optimal weight matrix, where $m$ is the dimension number of the state vector $x$, and $n$ denotes the node number of NNs.*

$$
A = \begin{bmatrix}
\omega_{11} & \omega_{12} & \cdots & \omega_{1m} \\
\omega_{21} & \omega_{22} & \cdots & \omega_{2m} \\
\vdots & \vdots & \ddots & \vdots \\
\omega_{n1} & \omega_{n2} & \cdots & \omega_{nm}
\end{bmatrix}
$$

**Lemma 2 ([47]).** *For any variable $\varrho$ and positive continuous function $\xi(t)$ with $t > 0$, we obtain*

$$
|\varrho| - \varrho \tanh \frac{\varrho}{\xi(t)} \leq \varphi \xi(t),
$$

*where the constant $\varphi > 0$.*

**Lemma 3.** *For $\forall a, b \geq 0$, and $p, q > 0$, satisfying $1/p + 1/q = 1$, the inequality holds as $ab \leq \frac{a^p}{p} + \frac{b^q}{q}$.*

### 2.2. Description of the Communication Topologies

The graph theory is introduced to illustrate the communication flow of the UAV formation system. We define a weighted directed graph $\mathcal{G} = (\mathcal{V}, \mathcal{E}, \mathcal{A})$, where $\mathcal{V} = \{v_i, i = 1, \ldots, n\}$ denotes the set of vertices, $\mathcal{E} \subseteq \mathcal{V} \times \mathcal{V}$ represents the set of edges, $\mathcal{A} = [a_{i,j}] \in \mathbb{R}^{n \times n}$ is the weighted adjacent matrix of the graph $\mathcal{G}$, with the non-negative entries $\{a_{i,j}\}$, where $a_{i,j} = 1$ for $(i, j) \in \mathcal{E}$ and $i \neq j$, while $a_{i,j} = 0$ for the remaining cases, which indicates that the $i$-th UAV receives information from the $j$-th UAV when $(i, j) \in \mathcal{E}$. For the $j$-th UAV, the set of its neighbors is denoted as $n_j = \{i \in \mathcal{V} | (i, j) \in \mathcal{E}\}$. We define $\mathcal{L} = \mathcal{D} - \mathcal{A}$ as the Laplacian matrix of the graph $\mathcal{G}$, where the diagonal matrix $\mathcal{D} = diag\left(d_1^{in}, \ldots, d_n^{in}\right)$, $d_i^{im} = \sum_{j=1}^n a_{i,j}$ is

defined as the indegree of vertex $p$. Here, we discuss the case that the graph is fixed and strongly connected during the control process, which means a path exists between any pair of two vertices.

Moreover, we set a fictitious leader as vertex 0; with the command consensus reference trajectory as $y_d$, the communication is established between at least one UAV and vertex 0. Thus, the directed graph is extended as $\bar{\mathcal{G}} = (\mathcal{V} \cup \{0\}, \bar{\mathcal{E}}, \bar{\mathcal{A}})$, where $\bar{\mathcal{E}}, \bar{\mathcal{A}}$ are respectively the extended edge set and the extended weighted adjacent matrix.

## 3. Main Results

In this section, the event-triggering mode is firstly introduced. The tracking controller for the desired trajectory is designed separately along the $x$ and $y$ axes, taking into account predefined performance constraints. Based on the given event-triggered mechanism, the yaw angle and velocity are then controlled using the aperiodic sampling mode. Finally, the flight control scheme for the attitude stabilization of each agent in the formation system is presented.

### 3.1. Event-Triggering Mode

The event-triggering mode is derived based on the following sampling errors of the control input:

$$\varsigma_r(t) = \tau_{d,i}(t) - \bar{\tau}_{d,i}, \tag{6}$$

$$\varsigma_u(t) = T_{d,i}(t) - \bar{T}_{d,i}, \tag{7}$$

where $\varsigma_r(t)$ and $\varsigma_u(t)$ denote the sampling error of the control input $\tau_{d,i}(t)$ and $T_{d,i}(t)$, respectively, for the yaw angle loop and velocity loop. By defining $\varsigma_r, \phi_r, \gamma_r, \bar{\gamma}_r$ as positive constants, $\bar{\gamma}_r > \gamma_r + \phi_r$, the triggering mechanism for $\tau_{d,i}(t)$ is then presented as follows:

$$\begin{cases} \bar{\tau}_{d,i} = \tau_{d,i}(t_k), t \in [t_k, t_{k+1}) \\ t_{k+1} = \inf\{t \in R, |\varsigma(t)| \geq \gamma_r \tanh|\tau_{d,i}| + \phi_r\}, \end{cases} \tag{8}$$

$t_k$ is the update time with $k$ being a positive integer. It can be shown that $\bar{\tau}_{d,i}$ will not change in $t \in [t_k, t_{k+1})$ and be updated from $\tau_{d,i}(t_k)$ to $\tau_{d,i}(t_{k+1})$ at $t = t_{k+1}$, $\{t_k, k \in \mathbb{N}\}$ are represented as the triggering time instants, and the first event occurs at $t_0 = 0$. $t_k - t_{k-1}$, which denotes the interval of two consecutive instants, is named as the inter-event time, where it is seen that the sampling error will be reset to 0 at each triggering instant. Similarly, by defining $\varsigma_u, \phi_u, \gamma_u, \bar{\gamma}_u$ as positive constants, $\bar{\gamma}_u > \gamma_u + \phi_u$, $\varsigma_u(t) = T_{d,i}(t) - \bar{T}_{d,i}$, the triggering mechanism of $T_{d,i}(t)$ is expressed as

$$\begin{cases} \bar{T}_{d,i} = T_{d,i}(t_k), t \in [t_k, t_{k+1}) \\ t_{k+1} = \inf\{t \in R, |\varsigma(t)| \geq \gamma_u \tanh|T_{d,i}| + \phi_u\}, \end{cases} \tag{9}$$

One also has that $\bar{T}_{d,i}$ will not change in $t \in [t_k, t_{k+1})$ and be updated from $T_{d,i}(t_k)$ to $T_{d,i}(t_{k+1})$ at $t = t_{k+1}$.

### 3.2. Trajectory-Tracking Controller Design with Predefined Performance Constraints

A controller for tracking the formation of UAVs is designed here, achieving the performance constraints derived from the Lyapunov candidate function, assuming that member $i$ of the formation is adjacent to member $j$. The goal of the controller is to obtain location $p_i$ closer to the vector $\boldsymbol{p_{i,d}} = [x_{i,d}, y_{i,d}]^{\mathrm{T}} = [x_j, y_j]^{\mathrm{T}} + [\Delta x_{i,j}, \Delta y_{i,j}]^{\mathrm{T}}$ representing the target value.

From Figure 1,

$$\begin{bmatrix} e_{x,i} \\ e_{y,i} \end{bmatrix} = \begin{bmatrix} \cos\psi_{ij,d} & -\sin\psi_{ij,d} \\ \sin\psi_{ij,d} & \cos\psi_{ij,d} \end{bmatrix}^T (\boldsymbol{p}_i - \boldsymbol{p_{i,d}}) \tag{10}$$

where $\psi_{ij,d} = \mathrm{atan}\,2\Big(y'_j(t) + \Delta x'_{i,j}(t), x'_j + \Delta y'_{i,j}\Big) \in [-\pi, \pi]$.

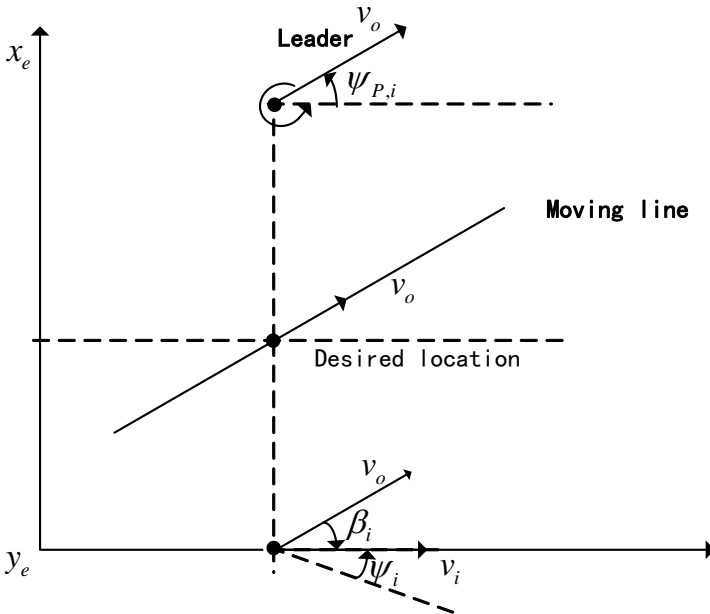

**Figure 1.** Schematic diagram of geometric relationship.

Obviously, the derivative of $e_{y,i}$ can be obtained as

$$\dot{e}_{y,i} = u_i \sin\Big(\psi_i - \psi_{ij,d}\Big) \tag{11}$$

Based on the vector field guidance law, the desired yaw angle $\psi_{d,i}$ is designed as

$$\psi_{d,i} = \psi_{ij,d} + \arcsin(-\frac{k_y b_{y,i}^2}{2u_i \pi e_{y,i}} \sin(\frac{\pi e_{y,i}^2}{b_{y,i}^2}) + \frac{\dot{b}_{y,i}}{u_i b_{y,i}} e_{y,i}) \tag{12}$$

where $k_y > 0$ is the guidance law parameter.

From Figure 1, the derivative of $e_{x,i}$ can be obtained as

$$\dot{e}_{x,i} = u_i \cos \beta_i - u_j \tag{13}$$

Then the desired velocity is designed as

$$u_{d,i} = \frac{1}{\cos \beta_i}(-\frac{k_v b_{x,i}^2}{2\pi e_{x,i}} \sin(\frac{\pi e_{x,i}^2}{b_{x,i}^2}) + \frac{\dot{b}_{x,i}}{b_{x,i}} e_{x,i} + u_j) \tag{14}$$

where $\beta_i = \psi_{ij,d} - \psi_i$, and the predefined performance constraints can be satisfied as $||e_{x,i}|| \le b_{x,i}, ||e_{y,i}|| \le b_{y,i}$ $b_{y,i}$ and $b_{x,i}$ are respectively the upper bound of $e_{y,i}$ and $e_{x,i}$. Detailing analysis is carried out via the derivation of the Lyapunov candidate function, see after from (A1) to (A4).

### 3.3. Yaw Angle Controller Design

For the formation member, we define the angle error as

$$e_{\psi,i} = \psi_i - \psi_{d,i} \tag{15}$$

To deal with the differential explosion of $\psi_{d,i}$, a command filter is introduced as

$$\dot{X}_1 = X_2$$
$$\dot{X}_2 = -2\zeta\omega_n X_2 - \omega_n^2(X_1 - \psi_{d,i})$$

where $\zeta$ and $\omega_n$ are filter parameters, $\psi_{d,i}$ is the input, $X_1$ is the estimation of $\psi_{d,i}$, and $X_2$ is the estimation of the derivative of $\psi_{d,i}$, $\hat{\dot{\psi}}_{d,i} = X_2$.

The estimate error of this command filter is defined as

$$\tilde{\dot{\psi}}_{d,i} = \dot{\psi}_{d,i} - \hat{\dot{\psi}}_{d,i} \tag{16}$$

To deal with the estimation error, an adaptive law is designed as

$$\dot{\hat{d}}_{\psi,i} = k_{1,\psi,i}\left(e_{\psi,i} - k_{2,\psi,i}\hat{d}_{\psi,i}\right) \tag{17}$$

with positive parameters $k_{1,\psi,i}$, $k_{2,\psi,i}$.

The adaptive tracking controller and the desired yaw angular velocity can be designed as

$$r_{d,i} = -k_{3,\psi,i}e_{\psi,i} - \hat{d}_{\psi,i} + \hat{\dot{\psi}}_{d,i} \tag{18}$$

Then define the tracking error of the yaw angular velocity as

$$e_{r,i} = r_i - r_{d,i} \tag{19}$$

To deal with the differential explosion of $r_{d,i}$, a command filter is introduced as

$$\dot{Y}_1 = Y_2$$
$$\dot{Y}_2 = -2\zeta\omega_n Y_2 - \omega_n^2(Y_1 - r_{d,i})$$

where $\zeta$ and $\omega_n$ are filter parameters, $r_{d,i}$ is the input, $Y_1$ is the estimation of $r_{d,i}$, and $Y_2$ is the estimation of the derivative of $r_{d,i}$, $\hat{\dot{\psi}}_{d,i} = Y_2$.

The estimate error of this command filter is defined as

$$\tilde{\dot{r}}_{d,i} = \dot{r}_{d,i} - \hat{\dot{r}}_{d,i} \tag{20}$$

Differentiating the tracking error of the yaw angular velocity,

$$\dot{e}_{r,i} = \dot{r}_i - \dot{r}_{d,i}$$
$$= \tau_i + \Delta\tau_i + d_{r,i} - \hat{\dot{r}}_{d,i} - \tilde{\dot{r}}_{d,i} \tag{21}$$

An RBFNN is employed to approximate unknown non-linear functions caused by the actuator fault. Then,

$$\dot{e}_{r,i} = r_i + A_{r,i}S_{r,i} + \varepsilon_{r,i} + d_{r,i} - \hat{\dot{r}}_{d,i} - \tilde{\dot{r}}_{d,i}$$
$$= r_i + A_{r,i}S_{r,i} - \hat{\dot{r}}_{d,i} + \tilde{d}_{r,i} \tag{22}$$

where $\tilde{d}_{r,i} = \varepsilon_{r,i} + d_{r,i} - \tilde{\dot{r}}_{d,i}$ are the total disturbances. Apparently, $\tilde{d}_{r,i}$ is bounded, satisfying $\tilde{d}_{r,i} \leq \bar{d}_{r,i}$. $\tilde{A}_{r,i} = A_{r,i} - \hat{A}_{r,i}$ and $A_{r,i} \leq \bar{A}_{r,i}$.

To deal with the unknown non-linear functions caused by the actuator fault, the adaptive law is designed as

$$\dot{\hat{A}}_{r,i} = k_{1,r,i}\left(S_{r,i}e_{r,i} - k_{2,r,i}\hat{A}_{r,i}\right) \tag{23}$$

with positive parameters $k_{1,r,i}$ and $k_{2,r,i}$.

To deal with the total disturbance, another adaptive law is designed as

$$\dot{\hat{d}}_{r,i} = k_{3,r,i}\left(e_{r,i} - k_{4,r,i}\hat{d}_{r,i}\right) \tag{24}$$

with positive parameters $k_{3,r,i}$, $k_{4,r,i}$.

The adaptive tracking controller can be designed as

$$\tau_{d,i} = -k_{5,r,i}e_{r,i} - \hat{d}_{r,i} - \hat{A}_{r,i}S_{r,i} + \dot{\hat{r}}_{d,i} - \bar{\gamma}_r \tanh\left(\frac{\bar{\gamma}_r e_{r,i}}{\xi_r}\right) \tag{25}$$

Adopting the event-triggering mode, it is known from (A9) that

$$|\tau_{d,i}(t) - \bar{\tau}_{d,i}| < \gamma_r \tanh|\bar{\tau}_{d,i}| + \phi_r$$
$$< \gamma_r + \phi_r$$

for $t \in [t_k, t_{k+1})$ such that $\bar{\tau}_{d,i}$ can be written as

$$\bar{\tau}_{d,i} = \tau_{d,i}(t) - \rho_r(t)(\gamma_r + \phi_r),$$

where $\rho_r(t) = \frac{\tau_{d,i}(t) - \bar{\tau}_{d,i}}{\gamma_r + \phi_r}$ is continuous and $|\rho_r(t)| < 1$.

### 3.4. Velocity Controller Design

For the formation member, we define the velocity error as

$$e_{u,i} = u_i - u_{d,i} \tag{26}$$

To deal with the differential explosion of $u_{d,i}$, a command filter is introduced as

$$\dot{Z}_1 = Z_2$$
$$\dot{Z}_2 = -2\zeta\omega_n Z_2 - \omega_n{}^2(Z_1 - u_{d,i})$$

where $\zeta$ and $\omega_n$ are filter parameters, $u_{d,i}$ is the input, $Z_1$ is the estimation of $u_{d,i}$, and $Z_2$ is the estimation of the derivative of $u_{d,i}$, $\hat{\dot{u}}_{d,i} = Z_2$.

The estimate error of this command filter is defined as

$$\tilde{u}_{d,i} = \dot{u}_{d,i} - \hat{\dot{u}}_{d,i} \tag{27}$$

Differentiating the tracking error of the velocity,

$$\dot{e}_{u,i} = \dot{u}_i - \dot{u}_{d,i}$$
$$= T_i + \Delta T_i + d_{u,i} - \hat{\dot{u}}_{d,i} - \tilde{u}_{d,i} \tag{28}$$

An RBFNN is employed to approximate unknown non-linear functions caused by the actuator fault. Then,

$$\dot{e}_{u,i} = u_i + A_{u,i}S_{u,i} + \varepsilon_{u,i} + d_{u,i} - \hat{\dot{u}}_{d,i} - \tilde{u}_{d,i}$$
$$= u_i + A_{u,i}S_{u,i} - \hat{\dot{u}}_{d,i} + \tilde{d}_{u,i} \tag{29}$$

where $\tilde{d}_{u,i} = \varepsilon_{u,i} + d_{u,i} - \tilde{u}_{d,i}$ are the total disturbances. Apparently, $\tilde{d}_{u,i}$ is bounded, satisfying $\tilde{d}_{u,i} \leq \bar{d}_{u,i}$. $\tilde{A}_{u,i} = A_{u,i} - \hat{A}_{u,i}$ and $A_{u,i} \leq \bar{A}_{u,i}$.

To deal with the unknown non-linear functions caused by the actuator fault, the adaptive law is designed as

$$\dot{\hat{A}}_{u,i} = k_{1,u,i}\big(S_{u,i}e_{u,i} - k_{2,u,i}\hat{A}_{u,i}\big) \tag{30}$$

with positive parameters $k_{1,u,i}$ and $k_{2,u,i}$.

To deal with the total disturbance, another adaptive law is designed as

$$\dot{\hat{d}}_{u,i} = k_{3,u,i}\left(e_{u,i} - k_{4,u,i}\hat{d}_{u,i}\right) \tag{31}$$

with positive parameters $k_{3,u,i}, k_{4,u,i}$.

The adaptive tracking controller can be designed as

$$T_{d,i} = -k_{5,u,i}e_{u,i} - \hat{d}_{u,i} - \hat{A}_{u,i}S_{u,i} + \hat{u}_{d,i} - \bar{\gamma}_u \tanh\left(\frac{\bar{\gamma}_u e_{u,i}}{\zeta_u}\right) \tag{32}$$

With the event-triggered mechanism, it is known from (A9) that

$$|T_{d,i}(t) - \bar{T}_{d,i}| < \gamma_u \tanh |\bar{T}_{d,i}| + \phi_u$$
$$< \gamma_u + \phi_u$$

for $t \in [T_k, T_{k+1})$ such that $\bar{T}_{d,i}$ can be written as

$$\bar{T}_{d,i} = T_{d,i}(t) - \rho_u(t)(\gamma_u + \phi_u),$$

where $\rho_u(t) = \frac{T_{d,i}(t) - \bar{T}_{d,i}}{\gamma_u + \phi_u}$ is continuous and $|\rho_u(t)| < 1$.

### 3.5. Zeno Behavior Analysis

In this subsection, the feasibility of the proposed event-triggering mode is analyzed, that is, the Zeno behavior is verified to be excluded. For the triggering strategy in the yaw angle loop, based on $\varsigma_r(t) = \tau_{d,i}(t) - \bar{\tau}_{d,i}$, one has:

$$\dot{\varsigma}_u(t) = \dot{\tau}_{d,i}(t) - \dot{\bar{\tau}}_{d,i}. \tag{33}$$

Owing to $\bar{\tau}_{d,i}$ being a constant for $t \in [t_k, t_{k+1})$, it is observed that

$$\dot{\varsigma}_u(t) = \dot{\tau}_{d,i}(t), \tag{34}$$

which implies $\dot{\varsigma}_u(t) \le |\dot{\tau}_{d,i}(t)|$.

According to (25), $\dot{\tau}_{d,i}(t)$ can be expressed as

$$\dot{\tau}_{d,i}(t) = -k_{5,r,i}\dot{e}_{r,i} - \dot{\hat{d}}_{r,i} - \hat{A}_{r,i}\dot{S}_{r,i} + \dot{\hat{r}}_{d,i} - \frac{\bar{\gamma}_r^2(\dot{e}_{r,i}\zeta_r + e_{r,i}\dot{\zeta}_r)}{\zeta_r^2 \cosh^2 \frac{\gamma_r e_{r,i}}{\zeta_r}}. \tag{35}$$

Obviously, $\dot{\tau}_{d,i}(t)$ is continuous and bounded:

$$|\dot{\tau}_{d,i}(t)| \le \Gamma, \tag{36}$$

where the constant $\Gamma > 0$.

Therefore, we obtain

$$\int_{T_k}^{T_{k+1}} \dot{\varsigma}_u(t)dt = \varsigma_u(T_{k+1}) - \varsigma_u(T_k)$$
$$= \gamma \tanh(|\bar{\tau}_{d,i}|) + \gamma_r - 0$$
$$\le \Gamma(T_{k+1} - T_k). \tag{37}$$

It means that

$$T_{k+1} - T_k \ge \frac{\gamma \tanh(|\bar{\tau}_{d,i}|) + \gamma_r}{\Gamma} \ge \frac{\gamma_r}{\Gamma}, \tag{38}$$

which reveals that the Zeno behavior is excluded for the triggering mode in the yaw angle loop.

Similarly, for the velocity loop, we can also obtain the boundedness of $\dot{T}_{d,i}(t)$ with the upper bound of $|\dot{T}_{d,i}(t)| \le \Xi$, where $\Xi > 0$. Therefore, the triggering interval $T_{k+1} - T_k$ for the velocity loop exists in the lower bound, which is given as

$$T_{k+1} - T_k \ge \frac{\gamma \tanh(|\bar{T}_{d,i}|) + \gamma_r}{\Xi} \ge \frac{\gamma_v}{\Xi}, \tag{39}$$

Thus, the Zeno behavior is completely excluded and the feasibility analysis is achieved.

**Remark 1.** *The entire flight control system of each agent in the formation system is actually included in the pitch angle control law, pitch rate control law, and the controller for the angle of attack in the longitude loop as well as the other controller to stabilize the attitude responses in the lateral loop. The above control laws, which are not mentioned in this paper, actually exist in the control system for each aircraft, which mainly refers to [17], and are seen as a general design for the fixed-wing aircraft.*

Additionally, Appendix A conducts stability analysis for each control axis.

## 4. Simulation Results

A formation model consisting of a leader and three followers is established for simulation purposes to test the effectiveness of the formation-tracking controller proposed in this study.

Four UAVs are created for simulated verification; one leader (AL) and three followers (AF1, AF2, and AF3) are selected based on the leader–follower technique. Figure 2 depicts the communication topology diagram between UAVs.

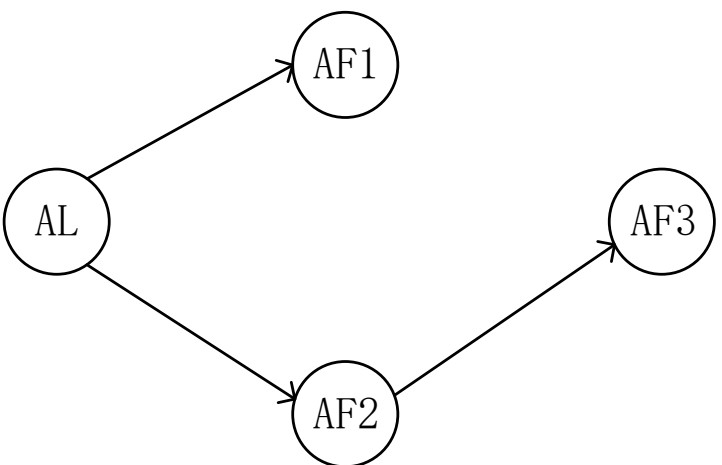

**Figure 2.** Communication topology diagram among UAV formations.

According to the directed communication topological graph between UAVs, that is, Figure 2, the Laplacian matrix can be obtained as:

$$L = \begin{bmatrix} 0 & 0 & 0 & 0 \\ -1 & 1 & 0 & 0 \\ -1 & 0 & 1 & 0 \\ -1 & 0 & 0 & 1 \end{bmatrix}$$

The assumptions for the simulation are as follows:
The external disturbances are set as

$$\begin{cases} d_u = 0.3\sin\left(0.1t + \frac{\pi}{6}\right) \\ d_r = 0.2\sin\left(0.2t + \frac{\pi}{4}\right) \end{cases}$$

The control inputs under fault conditions are set as

$$\begin{cases} T = T_c + p_u(t - T_c)[(l_u(t) - 1)T_c + \bar{T}_c] \\ \tau = \tau_c + p_r(t - T_c)[(l_r(t) - 1)\tau_c + \bar{\tau}_c] \end{cases}$$

in which $\boldsymbol{u} = [T, \tau]^{\mathrm{T}}$ are the real control inputs; $\boldsymbol{u}_c = [T_c, \tau_c]^{\mathrm{T}}$ are the control inputs calculated by the controller; $\boldsymbol{l} = [l_u(t), l_r(t)]^{\mathrm{T}}$ is the effectiveness parameter of the actuator, satisfying $0 < l_i(t) < 1 \quad (i = u, r)$; $\bar{\boldsymbol{u}}_c = [\bar{T}_c, \bar{\tau}_c]^{\mathrm{T}}$ denotes unknown inputs under fault conditions; and $\boldsymbol{p} = [p_u, p_r]^{\mathrm{T}}$ is the fault distribution with $p_i(t - T_c)(i = u, r)$, which is

$$p_i(t - T_c) = \begin{cases} 0 & \text{if } t < T_c \\ 1 - e^{-a(t-T_c)} & \text{if } t \geq T_c \end{cases}$$

where $a > 0$, $T_c$ is the instant when the actuator fault occurs.

Using the controller designs presented in Section 3, the following control parameters are selected:

The guidance law parameters of the followers are $k_{yi} = 1.01, k_{vi} = 1.1, i = 1, 2, 3$. The yaw controller parameters are $k_{\psi 1} = 10, k_{\psi 2} = 0.01, k_{\psi 3} = 1, k_{r1} = 1, k_{r2} = 0.01, k_{r3} = 3$. The velocity controller parameters are $k_{v1} = 1, k_{v2} = 0.01, k_{v3} = 0.1$. Meanwhile, in the neural network parameters, the number of neural network nodes is 301, and the width of the Gaussian function is 0.1. The adaptive law parameters are $k_{w\psi} = 8, k_{wv} = 18$. The error limit range is set as $x_b = 2 + 40e^{-0.2t}, y_b = 2 + 40e^{-0.2t}$.

Additionally, in this simulation, the expected time-varying trajectory of the leader is:

$$x_d = 20 \sin(t/10) + 20 \cos(t/5)$$
$$y_d = 20t$$
$$V_{xd} = 2 \cos(t/10) - 4 \sin(t/5)$$
$$V_{yd} = 20$$
$$\psi_d = a \tan 2\left(V_{yd}, V_{xd}\right)$$
$$V_d = \sqrt{V_{xd}^2 + V_{yd}^2}$$

The initial state of each follower and the relative expected position with the leader are shown in Table 1.

**Table 1.** Initial states of all UAVs.

| UAV (1,2,3) | Initial State $[x_0(m)\ y_0(m)\ \psi_0(\text{rad/s})\ v_0\text{m/s}\ \gamma_0(\text{rad})]^T$ | Desired Position with the Leader $[x_r(m)\ y_r(m)]^T$ |
|---|---|---|
| $P_1$ | $[-15.5\ -26.9\quad 0\quad 0\quad 1]^T$ | $[-40\ -40]^T$ |
| $P_2$ | $[-16.5\quad 50\quad \pi/6\quad 0\quad 1]^T$ | $[-40\ -40]^T$ |
| $P_3$ | $[-64.5\quad 20\quad \pi/3\ 0\ 1]^T$ | $[-80\ 0]^T$ |

Among them, the unknown disturbance received by each follower is $\psi_{id} = 0.001 \sin(t/10)$ and $v_{id} = 0.02 \sin(t/10)$.

In this paper, the sampling period is set as $h = 0.01$ s, and the whole sampling time is $T = 100$ s. Under the action of the controller proposed in this paper, four UAVs form a time-varying formation. The tracking effect of the formation trajectory is shown in Figure 3. In order to compare the performance of the proposed control algorithm in this paper, we employed the fixed-time sliding mode control algorithm based on event-triggered control proposed in reference [48] to achieve formation control. The specific results are shown in Figure 4.

Through the comparison of simulation results, it can be observed that the control algorithm proposed in this paper can accurately estimate and compensate for disturbances via RBFNN, effectively suppressing the adverse effects of disturbances on the precision of formation control. In contrast, the method provided in reference [48], while suppressing disturbances to some extent, exhibits a decline in controller performance when the desired reference trajectory undergoes abrupt changes, leading to oscillations in the formation flight trajectory of follower UAVs, as indicated by the marked position in the red box in Figure 4.

The tracking error results in the $X$ and $Y$ directions of the followers, which are shown in Figures 5 and 6.

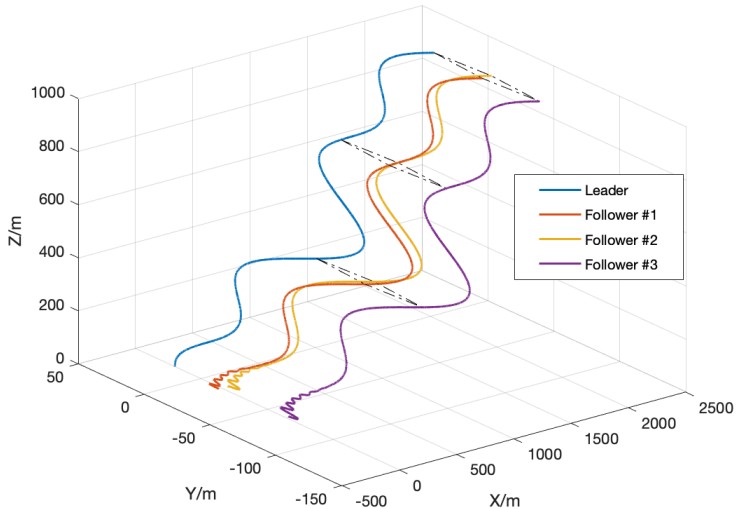

**Figure 3.** The control algorithm proposed in this paper.

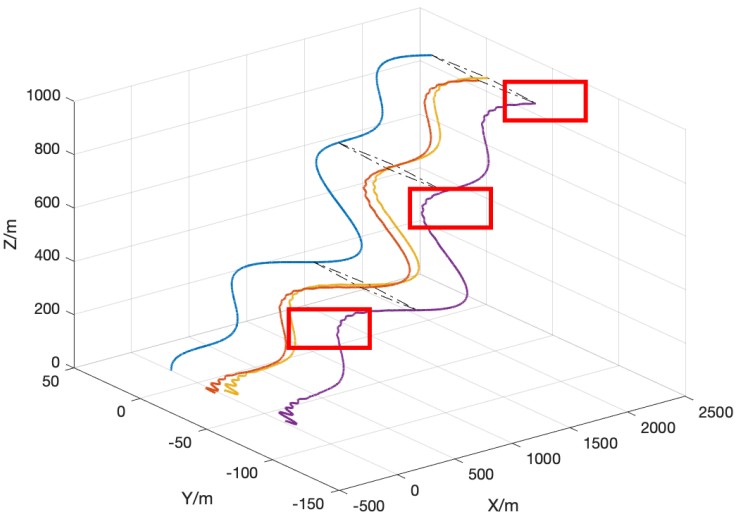

**Figure 4.** The control algorithm proposed in reference [48].

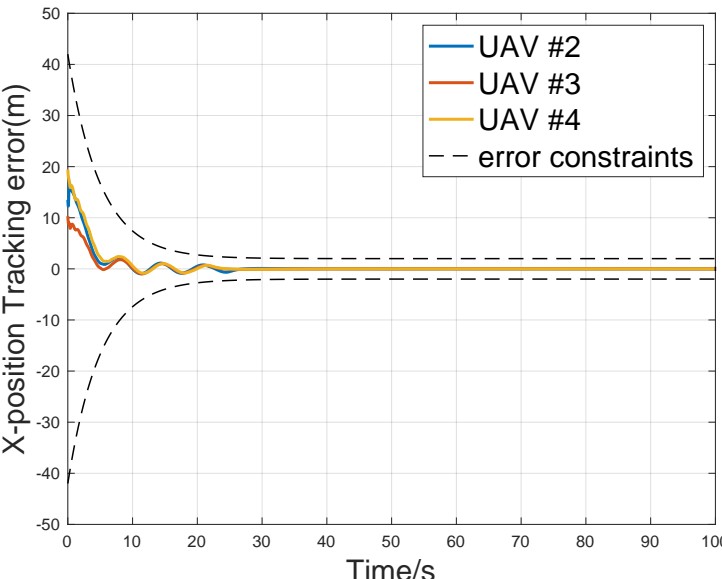

**Figure 5.** Leader -position tracking error.

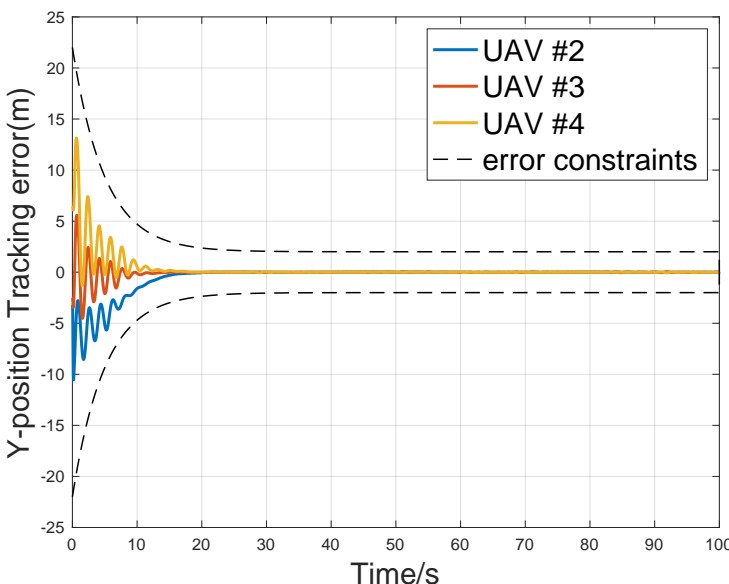

**Figure 6.** Follower-position tracking error.

It can be seen from the simulation results that the controller designed in this paper can complete the formation task in the presence of external disturbances and actuator failures, and the following error of the follower in the *X* and *Y* directions is always controlled within the tracking error. The follower's tracking error in the *X*-direction converges to zero around 28 s, and in the *Y*-direction, the tracking error converges to zero approximately at 18 s. The formation then achieves stable flight over the course of the mission.

During the formation process, the flight speed and heading angle of the leader and the follower change as shown in Figures 7 and 8.

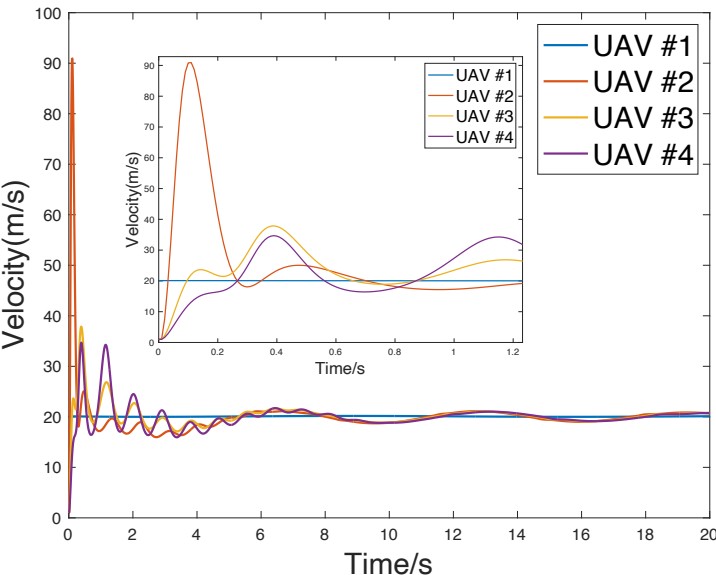

**Figure 7.** Velocity–time graph.

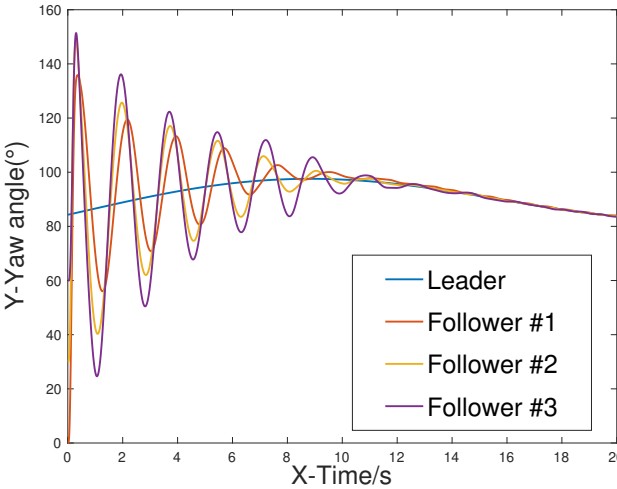

**Figure 8.** Variation curve of heading angle.

It can be seen from the figures that the final flight speed and heading angle of the follower tend to be consistent with the leader, which ensures that the followers are finally stable at the desired formation position. In comparison to other UAVs, UAV 2 initially has the furthest distance from the desired position. Therefore, to ensure it catches up with the expected aircraft trajectory as quickly as possible, at the beginning of the simulation, UAV 2 flies towards the desired trajectory at a faster speed. Approximately 8 s later, the UAV's speed matches the desired speed. The heading angle of the UAV aligns with the expected heading angle around 14 s.

The effect of RBFNN on the estimation of the external uncertain disturbance is shown in Figures 9 and 10.

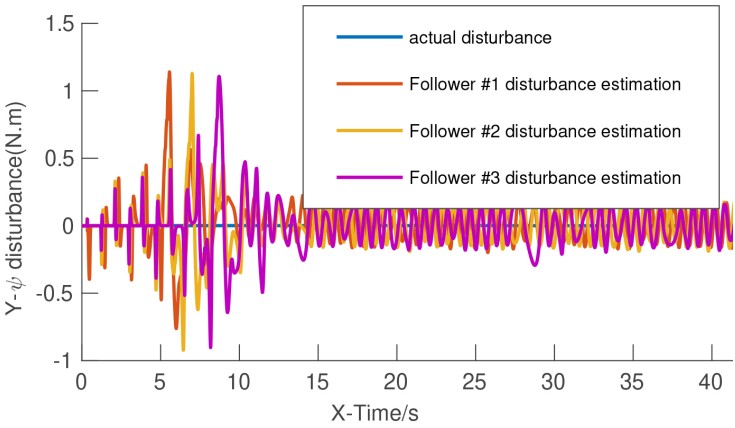

**Figure 9.** The estimated effect of the disturbance.

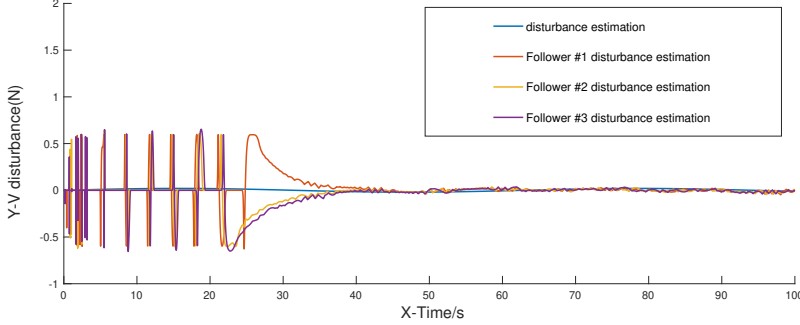

**Figure 10.** The estimated effect of the disturbance.

From the above results, it can be seen that the RBFNN neural network proposed in this paper can effectively estimate and compensate for the external uncertain disturbance.

Figures 11 and 12 depict the changing curve of the actual output force (torque) of the follower actuator.

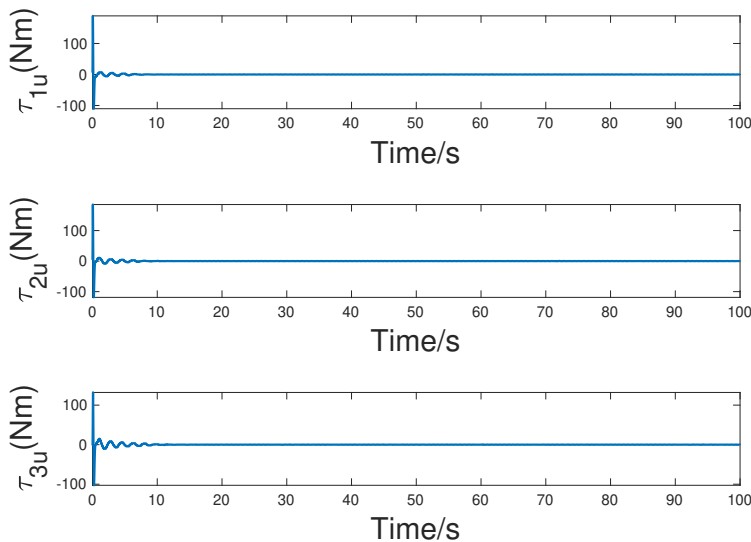

**Figure 11.** Output force (torque) variation curve of actuator for leader.

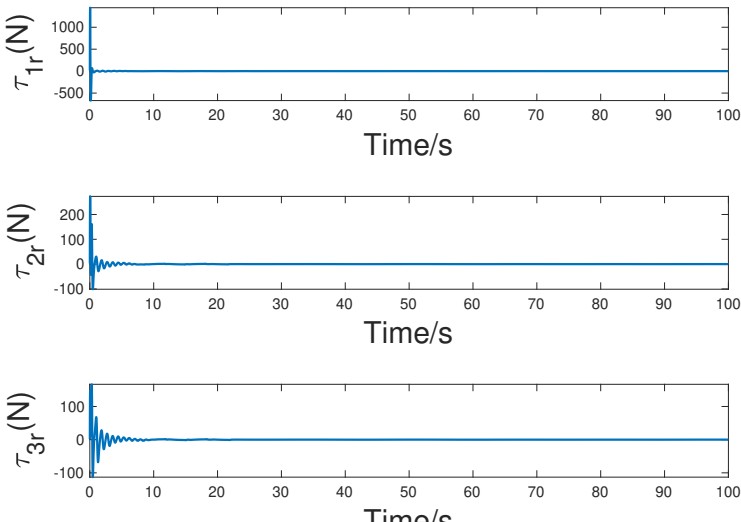

**Figure 12.** Output force (torque) variation curve of actuator for follower.

It can be seen from the simulation results that the actual output force of the actuator can be stabilized within 10 s. The method designed in this paper can effectively eliminate the influence of the actuator failure on the actual output force (torque) and ensure the formation-tracking effect of the followers.

A statistical analysis of the event-triggered intervals for the heading controller and speed controller throughout the entire formation process is presented in Table 2. The minimum trigger interval for the event-triggered controllers is 0.01 s (equivalent to one simulation timestep), while the maximum trigger interval is in the order of seconds. The average trigger interval is approximately 0.11 s. Compared to the traditional time-triggered controllers with a 0.01 s trigger interval, the event-triggered mechanism effectively reduces the update frequency of the controllers.

**Table 2.** Event-triggering times of each UAV.

| UAV No. | Heading Angle Control Time Interval | | | Velocity Control Time Interval | | |
|---|---|---|---|---|---|---|
| | Minimum Time (s) | Maximum Time (s) | Average Time (s) | Minimum Time (s) | Maximum Time (s) | Average Time (s) |
| UAV 1 | 0.010 | 0.970 | 0.124 | 0.010 | 2.390 | 0.118 |
| UAV 2 | 0.010 | 1.850 | 0.120 | 0.010 | 2.040 | 0113 |
| UAV 3 | 0.010 | 1.130 | 0.121 | 0.010 | 3.780 | 0.097 |

As shown in Figures 13 and 14, they are the event triggering times of the heading angle control and velocity control, respectively. In the initial stage of simulation, there is a significant error in the speed and heading angles of the three UAVs compared to the desired values, leading to frequent activations of the controllers, as indicated by the dense points in the graph. To ensure that the UAVs promptly track the desired speed and heading angles, the controllers undergo frequent updates. As analyzed in the preceding text, approximately after 8 s, the speed of the UAVs successfully aligns with the desired speed, and thereafter, the trigger frequency of the speed controller begins to decrease. Similarly, after approximately 14 s, the heading angle of the UAVs matches the desired heading angle, and subsequently, the trigger frequency of the heading controller gradually decreases.

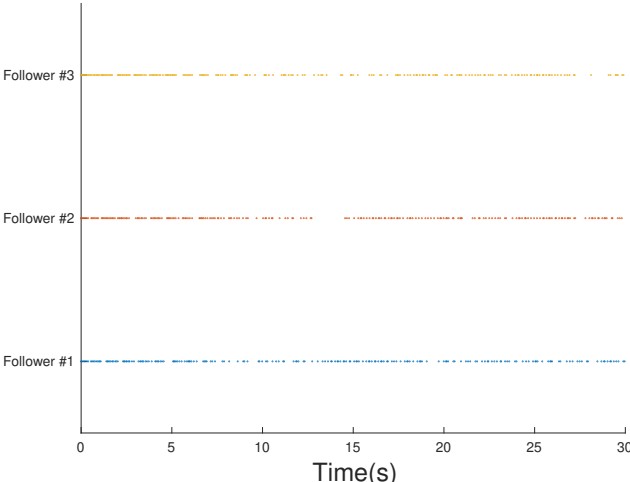

**Figure 13.** The event triggering times of the heading angle control.

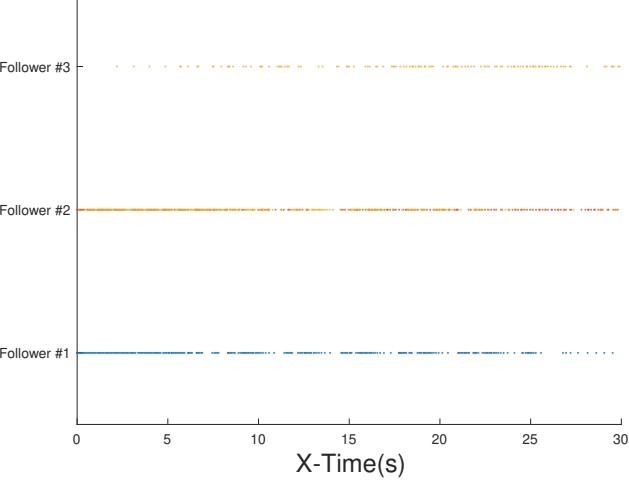

**Figure 14.** The event triggering times of the velocity control.

As shown in Figure 15, the comparison between the trigger counts of the proposed event-triggered controllers and the total simulation steps is presented in this paper. The yellow portion in the graph represents the total simulation steps, which correspond to the trigger counts of the time-triggered controllers. The red and blue sections, respectively, indicate the trigger counts of the event-triggered heading angle controller and speed controller.

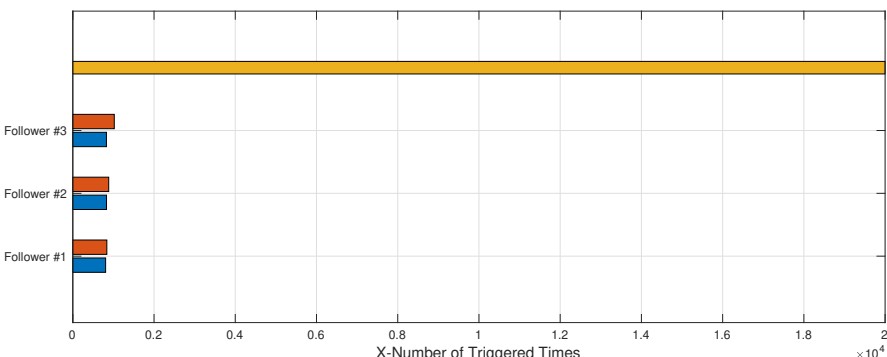

**Figure 15.** Contrast of triggered times.

As shown in Figure 15, the event-triggered controllers proposed in this paper not only effectively accomplish time-varying formation-tracking flight but also significantly reduce the demands on computational resources and communication bandwidth. Specifically, the computational resource and communication bandwidth requirements for UAVs 1, 2, and 3 account for only 8.2%, 8.6%, and 9.2%, respectively, compared to the time-triggered controllers. Therefore, under the same conditions of flight control computer hardware and communication resources, the event-triggered controllers proposed in this paper can more efficiently meet the formation requirements of a larger number of UAVs compared to the time-triggered controllers. This results in a substantial improvement in the utilization efficiency of flight control computers.

## 5. Conclusions

In this study, we propose an event-triggered fault-tolerant time-varying formation control with a predefined tracking performance for multiple UAVs. This method considers predefined tracking performance, which ensures that the convergence error remains within a set bounded limit. We design the guidance law for formation tracking and set the tracking error performance using a Lyapunov function. Subsequently, we design an adaptive tracking controller that includes a neural network and adaptive law. The formation-tracking control is implemented even in the presence of actuator faults and external disturbances. We also incorporate event-triggered control to further enhance the efficiency of the controller's solution. Finally, we conduct simulation experiments under various conditions to demonstrate that our designed controller can still converge the tracking error within our predefined range, even in the presence of external disturbances and internal actuator failures.

Future work can involve experimental validation of the formation-tracking controller to assess its effectiveness in real-world scenarios. This includes testing the system under different environmental conditions, as well as with different types of disturbances and faults, to verify its robustness and fault tolerance.

In response to the issue of communication delays among UAVs and between UAVs and ground stations in real-flight environments, subsequent research will employ effective methods for compensation. For instance, a robust fault-tolerant tracking control scheme based on a fixed-time disturbance observer is proposed in reference [49]. This scheme combines Padé approximation and intermediate variable techniques, reducing the complexity of studying quadcopter UAV systems with input delays. The design of the fixed-time disturbance observer successfully eliminates the effects of compounded disturbances, offer-

ing a new research direction for addressing communication delay issues. Furthermore, to enhance the convergence speed of the controller designed in this paper, reference [50] introduces a novel non-singular fast terminal sliding mode surface. The design of this surface aims to avoid singularities and improve the performance of the controller while achieving a faster convergence speed. This approach is insightful for improving the convergence speed of the controller proposed in this paper, providing beneficial insights for enhancing the system performance.

**Author Contributions:** Conceptualization, Z.M. and H.G.; methodology, Z.M. and H.G.; software, Z.M.; validation, Z.M.; formal analysis, Z.M. and X.W.; investigation, Z.M.; resources, H.G.; data curation, H.G.; writing—original draft preparation, Z.M.; writing—review and editing, X.W.; visualization, Z.M.; supervision, H.G.; project administration, H.G. All authors have read and agreed to the published version of the manuscript.

**Funding:** This research  received no external funding.

**Institutional Review Board Statement:** Not applicable.

**Informed Consent Statement:** Not applicable.

**Data Availability Statement:** The original contributions presented in the study are included in the article material, further inquiries can be directed to the corresponding author.

**Conflicts of Interest:** The authors declare no conflicts of interest.

## Appendix A

In this section, the stability analysis is carried out on each control axis. The design of controllers is typically analyzed for stability using Lyapunov functions. Refs. [51,52] conducted stability analysis of controllers in adaptive schemes using non-quadratic Lyapunov functions. They considered a non-quadratic Lyapunov function denoted as $\alpha$, and when $\alpha = 1$, it reduces to a quadratic Lyapunov function. Through analytical discussions on the impact of different $\alpha$ values on system response, it was concluded that using multiple standard $\alpha$ values might result in faster responses compared to $\alpha = 1$. The adaptive law in this paper is designed for a second-order system, and therefore, quadratic Lyapunov functions are employed for stability analysis. Specifically, the predefined performance of the tracking error is proposed by the design of the proposed Lyapunov candidate function.

A Lyapunov function condition for the constrained tracking performance on the $y$ axis is given as

$$V_1 = \frac{b_{y,i}^2}{\pi} \tan\left(\frac{\pi e_{y,i}^2}{2b_{y,i}^2}\right) \tag{A1}$$

in which $e_{y,i}$ is the shortest distance between the real position $\boldsymbol{p}_i$ of the UAV $i$ and the moving line of the desired position $\boldsymbol{p}_{i,d}$.

Differentiating (11),

$$\dot{V}_1 = \frac{e_{y,i}\dot{e}_{y,i}}{\cos^2\left(\frac{\pi e_{y,i}^2}{2b_{y,i}^2}\right)} + \frac{2b_{y,i}\dot{b}_{y,i}}{\pi}\tan\left(\frac{\pi e_{y,i}^2}{2b_{y,i}^2}\right) - \left(\frac{\dot{b}_{y,i}}{b_{y,i}}\right)\frac{e_{y,i}^2}{\cos^2\left(\frac{\pi e_{y,i}^2}{2b_{y,i}^2}\right)} \tag{A2}$$

Defining $k_b = \sup\left|\frac{\dot{b}_{y,i}}{b_{y,i}}\right|$, according to (10), (A2), and (12), (A1) can be obtained as

$$\dot{V}_1 = \frac{y_{e,i}\dot{y}_{e,i}}{\cos^2\left(\frac{\pi y_{e,i}^2}{2b_{y,i}^2}\right)} + \frac{2b_{y,i}\dot{b}_{y,i}}{\pi}\tan\left(\frac{\pi y_{e,i}^2}{2b_{y,i}^2}\right) - \left(\frac{\dot{b}_{y,ic}}{b_{y,i}}\right)\frac{y_{e,i}^2}{\cos^2\left(\frac{\pi y_{e,i}^2}{2b_{y,i}^2}\right)}$$

$$\leq \frac{y_{e,i}u_i\sin\left(\psi_i - \psi_{ij,d}\right)}{\cos^2\left(\frac{\pi y_{e,i}^2}{2b_{y,i}^2}\right)} - \left(\frac{\dot{b}_{y,ic}}{b_{y,i}}\right)\frac{y_{e,i}^2}{\cos^2\left(\frac{\pi y_{e,i}^2}{2b_{y,i}^2}\right)} + \frac{2k_b b_{y,i}^2}{\pi}\tan\left(\frac{\pi y_{e,i}^2}{2b_{y,i}^2}\right)$$

$$\leq -(k_d - 2k_b)\frac{b_{y,i}^2}{\pi}\tan\left(\frac{\pi y_{e,i}^2}{2b_{y,i}^2}\right) \tag{A3}$$

Choosing $k_d > 2k_b$,

$$\dot{V}_1 \leq -\sigma_1 V_1 \tag{A4}$$

where $\sigma_1 = k_d - 2k_b$.

Similarly, to achieve the predefined tracking performance on the $x$ axis, the following Lyapunov function condition is given as

$$V_2 = \frac{b_{x,i}^2}{\pi}\tan\left(\frac{\pi e_{x,i}^2}{2b_{x,i}^2}\right) \tag{A5}$$

Differentiating (13),

$$\dot{V}_2 = \frac{x_{e,i}\dot{x}_{e,i}}{\cos^2\left(\frac{\pi x_{e,i}^2}{2b_{x,i}^2}\right)} + \frac{2b_{x,i}\dot{b}_{x,i}}{\pi}\tan\left(\frac{\pi x_{e,i}^2}{2b_{x,i}^2}\right) - \left(\frac{\dot{b}_{x,ic}}{b_{x,i}}\right)\frac{x_{e,i}^2}{\cos^2\left(\frac{\pi x_{e,i}^2}{2b_{x,i}^2}\right)}$$

$$\leq \frac{x_{e,i}\left(u_i\cos\beta_i - u_j\right)}{\cos^2\left(\frac{\pi x_{e,i}^2}{2b_{x,i}^2}\right)} - \left(\frac{\dot{b}_{x,i}}{b_{x,i}}\right)\frac{x_{e,i}^2}{\cos^2\left(\frac{\pi x_{e,i}^2}{2b_{x,i}^2}\right)} + \frac{2k_c b_{x,i}^2}{\pi}\tan\left(\frac{\pi x_{e,i}^2}{2b_{x,i}^2}\right)$$

$$\leq -(k_v - 2k_c)\frac{b_{x,i}^2}{\pi}\tan\left(\frac{\pi x_{e,i}^2}{2b_{x,i}^2}\right) \tag{A6}$$

where $k_c = \sup\left|\frac{\dot{b}_{x,i}}{b_{x,i}}\right|$.

Choosing $k_v > 2k_c$,

$$\dot{V}_2 \leq -\sigma_2 V_2 \tag{A7}$$

where $\sigma_2 = k_v - 2k_c$, which means that the Lyapunov candidate function $V_2$ is stable.

For the yaw angle control, consider the following Lyapunov candidate

$$V_3 = \frac{1}{2}e_{\psi,i}^2 + \frac{1}{2}k_{1,\psi,i}^{-1}\tilde{d}_{\psi,i}^2 + \frac{1}{2}e_{r,i}^2 + \frac{1}{2}k_{3,r,i}^{-1}\tilde{d}_{r,i}^2 + \frac{1}{2}k_{1,r,i}^{-1}\tilde{A}_{r,i}^2 \tag{A8}$$

Then

$$\dot{V}_3 = e_{\psi,i}\dot{e}_{\psi,i} + \hat{d}_{\psi,i}\dot{\hat{d}}_{\psi,i} + e_{r,i}\dot{e}_{r,i} + \hat{d}_{r,i}\dot{\hat{d}}_{r,i} + \tilde{A}_{r,i}\dot{\tilde{A}}_{r,i}$$

$$\leq -k_{3,\psi,i}e_{\psi,i}^2 - k_{2,\psi,i}\tilde{d}_{\psi,i}^2 + \|e_{\psi,i}\|\|\bar{d}_{\psi,i}\|$$

$$- k_{5,r,i}e_{r,i}^2 - k_{4,r,i}\tilde{d}_{r,i}^2 + \|e_{r,i}\|\|\bar{d}_{r,i}\| + k_{2,r,i}\tilde{A}_{r,i}\hat{A}_{r,i}$$

$$- e_{r,i}\left(\rho_r(t)(\gamma_r + \phi_r) + \bar{\gamma}_r\tanh\left(\frac{\bar{\gamma}_r e_{r,i}}{\zeta_r}\right)\right) \tag{A9}$$

Based on Lemma 2, it follows that

$$-e_{r,i}\left(\rho_r(t)(\gamma_r + \phi_r) + \gamma_r \tanh\left(\frac{\bar{\gamma}_r e_{r,i}}{\zeta_r}\right)\right) \leq \left(|\bar{\gamma}_r e_{r,i}| - \bar{\gamma}_r e_{r,i} \tanh\left(\frac{\bar{\gamma}_r e_{r,i}}{\zeta_r}\right)\right)$$
$$\leq \varphi_r \zeta_r.$$

where $\varphi_r \zeta_r > 0$ is a positive constant.

Then

$$\begin{aligned}
\dot{V}_3 \leq & - k_{3,\psi,i} e_{\psi,i}^2 - k_{2,\psi,i}\hat{d}_{\psi,i}^2 + \|e_{\psi,i}\|\|\bar{d}_{\psi,i}\| \\
& - k_{5,r,i} e_{r,i}^2 - k_{4,r,i}\hat{d}_{r,i}^2 + \|e_{r,i}\|\|\bar{d}_{r,i}\| \\
& + k_{2,r,i}\|\tilde{A}_{r,i}\|(\bar{A}_{r,i} - \|\tilde{A}_{r,i}\|) + \varphi_r \zeta_r
\end{aligned} \tag{A10}$$

where $\|A_{r,i}\| \leq \bar{A}_{r,i}$.

Using Lemma 3,

$$\|\tilde{A}_{r,i}\|(\bar{A}_{r,i} - \|\tilde{A}_{r,i}\|) \leq -\frac{1}{2}\|\tilde{A}_{r,i}\|^2 + \frac{1}{2}\bar{A}_{r,i}^2 \tag{A11}$$

$$\|e_{\psi,i}\|\|\bar{d}_{\psi,i}\| \leq \frac{1}{2}\|e_{\psi,i}\|^2 + \frac{1}{2}\bar{d}_{\psi,i}^2 \tag{A12}$$

$$\|e_{r,i}\|\|\bar{d}_{r,i}\| \leq \frac{1}{2}\|e_{r,i}\|^2 + \frac{1}{2}\bar{d}_{r,i}^2 \tag{A13}$$

Then

$$\dot{V}_3 \leq - (k_{3,\psi,i} - \frac{1}{2})e_{\psi,i}^2 - k_{2,\psi,i}\hat{d}_{\psi,i}^2 - (k_{5,r,i} - \frac{1}{2})e_{r,i}^2 - k_{4,r,i}\hat{d}_{r,i}^2 - \frac{1}{2}k_{2,r,i}\|\tilde{A}_{r,i}\|^2 \tag{A14}$$

$$+ \frac{1}{2}\bar{d}_{\psi,i}^2 + \frac{1}{2}\bar{d}_{r,i}^2 + k_{2,r,i}\frac{1}{2}\|\bar{A}_{r,i}\|^2 + \varphi_r \zeta_r \tag{A15}$$

The conclusion can be given that $\dot{V}_3 \leq -\sigma_3 V_3 + \zeta_1$ in which $\sigma_3 = \min\{k_{3,\psi,i} - \frac{1}{2}, k_{2,\psi,i}, k_{5,r,i} - \frac{1}{2}, k_{4,r,i}, k_{2,r,i}\} > 0$, $\zeta_1 = \frac{1}{2}\bar{d}_{r,i}^2 + k_{2,r,i}\frac{1}{2}\|\bar{A}_{r,i}\|^2 + \varphi_r \zeta_r > 0$.

The Lyapunov function of the velocity control loop is given as

$$V_4 = \frac{1}{2}e_{u,i}^2 + \frac{1}{2}k_{3,u,i}^{-1}\hat{d}_{u,i}^2 + \frac{1}{2}k_{1,u,i}^{-1}\tilde{A}_{u,i}^2 \tag{A16}$$

Then

$$\begin{aligned}
\dot{V}_4 =& e_{u,i}\dot{e}_{u,i} + \hat{d}_{u,i}\dot{\hat{d}}_{u,i} + \tilde{A}_{u,i}\dot{\tilde{A}}_{u,i} \\
\leq& - k_{5,r,i} e_{u,i}^2 - k_{4,r,i}\hat{d}_{u,i}^2 + e_{u,i}\bar{d}_{u,i} + k_{2,r,i}\tilde{A}_{u,i}\hat{A}_{u,i} \\
& - e_{u,i}\left(\rho_u(t)(\gamma_u + \phi_u) + \bar{\gamma}_u \tanh\left(\frac{\bar{\gamma}_u e_{u,i}}{\zeta_u}\right)\right)
\end{aligned} \tag{A17}$$

Based on Lemma 2, it follows that

$$-e_{u,i}\left(\rho_u(t)(\gamma_u + \phi_u) + \gamma_u \tanh\left(\frac{\bar{\gamma}_u e_{u,i}}{\zeta_u}\right)\right) \leq \left(|\bar{\gamma}_u e_{u,i}| - \bar{\gamma}_u e_{u,i} \tanh\left(\frac{\bar{\gamma}_u e_{u,i}}{\zeta_u}\right)\right)$$
$$\leq \varphi_u \zeta_u.$$

where $\varphi_u \zeta_u > 0$ is a positive constant.

Then

$$\dot{V}_4 \leq - k_{5,r,i} e_{u,i}^2 - k_{4,r,i}\hat{d}_{u,i}^2 + e_{u,i}\bar{d}_{u,i} + k_{2,r,i}\|\tilde{A}_{u,i}\|(\bar{A}_{u,i} - \|\tilde{A}_{u,i}\|) + \varphi_u \zeta_u \tag{A18}$$

where $\|A_{u,i}\| \leq \bar{A}_{u,i}$.

Using Lemma 3,

$$\left\| \tilde{A}_{u,i} \right\| (\bar{A}_{u,i} - \left\| \tilde{A}_{u,i} \right\|) \leq -\frac{1}{2} \left\| \tilde{A}_{u,i} \right\|^2 + \frac{1}{2} \bar{A}_{u,i}^2 \tag{A19}$$

$$e_{u,i} \bar{d}_{u,i} \leq \frac{1}{2} \left\| e_{u,i} \right\|^2 + \frac{1}{2} \bar{d}_{u,i}^2 \tag{A20}$$

Then

$$\dot{V}_4 \leq -(k_{5,r,i} - \frac{1}{2})e_{u,i}^2 - k_{4,r,i}\tilde{d}_{u,i}^2 - \frac{1}{2}k_{2,r,i}\left\| \tilde{A}_{u,i} \right\|^2 + \frac{1}{2}\bar{d}_{u,i}^2 + k_{2,r,i}\frac{1}{2}\left\| \bar{A}_{u,i} \right\|^2 + \varphi_u \xi_u \tag{A21}$$

The conclusion can be given that $\dot{V}_4 \leq -\sigma_4 V_4 + \zeta_2$ in which $\sigma_4 = \min\{k_{5,u,i} - \frac{1}{2}, k_{4,u,i}, k_{2,u,i}\} > 0$, $\zeta_2 = \frac{1}{2}\bar{d}_{u,i}^2 + k_{2,r,i}\frac{1}{2}\left\| \bar{A}_{u,i} \right\|^2 + \varphi_u \xi_u > 0$.

Define the Lyapunov function for the entire control system as

$$\begin{aligned}
V =& V_1 + V_2 + V_3 + V_4 \\
=& \frac{b_{x,i}^2}{\pi} \tan\left( \frac{\pi e_{x,i}^2}{2b_{x,i}^2} \right) + \frac{b_{y,i}^2}{\pi} \tan\left( \frac{\pi e_{y,i}^2}{2b_{y,i}^2} \right) + \frac{1}{2}e_{\psi,i}^2 \\
&+ \frac{1}{2}k_{1,\psi,i}^{-1}\tilde{d}_{\psi,i}^2 + \frac{1}{2}e_{r,i}^2 + \frac{1}{2}k_{3,r,i}^{-1}\tilde{d}_{r,i}^2 + \frac{1}{2}k_{1,r,i}^{-1}\tilde{A}_{r,i}^2 \\
&+ \frac{1}{2}e_{u,i}^2 + \frac{1}{2}k_{3,u,i}^{-1}\tilde{d}_{u,i}^2 + \frac{1}{2}k_{1,u,i}^{-1}\tilde{A}_{u,i}^2
\end{aligned} \tag{A22}$$

The conclusion can be given that $\dot{V} \leq -\sigma V + \zeta$ in which $\sigma = \min\{k_d - 2k_b, k_v - 2k_c, k_{3,\psi,i} - \frac{1}{2}, k_{2,\psi,i}, k_{5,r,i} - \frac{1}{2}, k_{4,r,i}, k_{2,r,i}, k_{5,u,i} - \frac{1}{2}, k_{4,u,i}, k_{2,u,i}\} > 0$, $\zeta_1 = \frac{1}{2}\bar{d}_{u,i}^2 + k_{2,u,i}\frac{1}{2}\left\| \bar{A}_{u,i} \right\|^2 + \frac{1}{2}\bar{d}_{\psi,i}^2 + \frac{1}{2}\bar{d}_{r,i}^2 + k_{2,r,i}\frac{1}{2}\left\| \bar{A}_{r,i} \right\|^2 > 0$.

Selecting parameters as $k_d > 2k_b$, $k_v > 2k_c$, $k_{5,\psi,i} > \frac{1}{2}$, $k_{5,u,i} > \frac{1}{2}$, $k_{2,\psi,i}$, $k_{2,u,i}$, $k_{4,u,i}$, $k_{2,r,i}$, $k_{4,r,i}$ and integrating (A22), we can obtain $V \leq \left( V(0) - \frac{\zeta}{\sigma} \right)e^{-\sigma t} + \frac{\zeta}{\sigma}$. The conclusion can be drawn that $V$ is bounded. Moreover, $\frac{b_{x,i}^2}{\pi}\tan\left( \frac{\pi e_{x,i}^2}{2b_{x,i}^2} \right) \leq V \leq \left( V(0) - \frac{\zeta}{\sigma} \right)e^{-\sigma t} + \frac{\zeta}{\sigma}$, $\frac{b_{y,i}^2}{\pi}\tan\left( \frac{\pi e_{y,i}^2}{2b_{y,i}^2} \right) \leq V \leq \left( V(0) - \frac{\zeta}{\sigma} \right)e^{-\sigma t} + \frac{\zeta}{\sigma}$, which means $e_{x,i}^2 \leq \frac{2b_{x,i}^2}{\pi}\tan^{-1}\left( \frac{\pi}{b_{x,i}^2}\left( \left( V(0) - \frac{\zeta}{\sigma} \right)e^{-\sigma t} + \frac{\zeta}{\sigma} \right) \right) < b_{x,i}^2$, $e_{y,i}^2 \leq \frac{2b_{y,i}^2}{\pi}\tan^{-1}\left( \frac{\pi}{b_{y,i}^2}\left( \left( V(0) - \frac{\zeta}{\sigma} \right)e^{-\sigma t} + \frac{\zeta}{\sigma} \right) \right) < b_{y,i}^2$.

Therefore, the conclusion can be drawn that $e_{x,i}$, $e_{y,i}$ are bounded as $|e_{x,i}| < |b_{x,i}|$, $|e_{y,i}| < |b_{y,i}|$ and can be eliminated to a small neighborhood around zero.

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
