# Peer review of "Fault-Tolerant Event-Triggrred Control for Multiple UAVs with Predefined Tracking Performance"

_drones, doi:10.3390/drones8010025_

Round 1
Reviewer 1 Report
Comments and Suggestions for Authors
This paper develops an event-triggering fault-tolerant time-varying formation control 1 for multiple unmanned aerial vehicles with predefined tracking performance. A complete theoretical proof is given and the validity of the proposed control scheme is given based on simulation. This study is written well, but the following questions need to be solved as: (1)As for equation (1), the definitions of some variables and parameters are missing; (2)Please explain why the torque in Figure 11 is suddenly so large at the beginning; (3)More interpretation and analysis is needed regarding the simulation results; (4)An adaptive NN control is designed in this study, but it would consume a lot of computing resources.Recently, some low computation schemes are proposed such as observer-based adaptive fuzzy finite-time attitude control for quadrotor UAVs, and antisaturation fixed-time attitude tracking control based low-computation learning for uncertain quadrotor UAVs with external disturbances,authors are advised to discuss these works in the introduction section; (6)Although many factors are considered in this article, But like input delay and fast convergence such as works in antisaturation adaptive fixed-time sliding mode controller design to achieve faster convergence rate and its application, and fixed-time disturbance observer-based robust fault-tolerant tracking control for uncertain quadrotor UAV subject to input delay, can improve the system performance. The authors can discuss them as one of the future research methods in the conclusion section; and (7)Authors are advised to discuss these works in the introductory chapters.
Reviewer 2 Report
Comments and Suggestions for Authors
Dear Authors,
the article titled "Event-triggered Fault-tolerant Time-varying Formation Control for Multiple UAVs with Predefined Tracking Performance" requires changes, especially in the first part of the work.
Authors begin descriptions with a chapter usually describing related works. The reader is not introduced to the concept that the authors want to present. The reader receives a large list of articles in the area of UAV swarm control, but has a lot of trouble checking what elements will be modified by the authors. In the Introduction chapter, the authors should refer to the articles and models that they will develop.
It is worth posting further discussion on Related Works. This will significantly organize the article and increase its readability.
I think it's a good idea to introduce aligning equations to the left side. It will be easier to read.
It would also be good to have a table with definitions of variables and parameters at the end of the text.
Section 3.5 analyzes the stability of the algorithm. Due to the complexity of the proof and the fact that skipping this section by the reader does not cause difficulties in analyzing the results, I propose to introduce the section as an appendix to the main text.
Question 1:
In Section 2.2 the graph is defined. Is this graph in the form of MST (Minimum Spanning Tree)?
Question 2:
You define four positive constatnt in line 194 (ξu, φu, γu, ¯γu) but I can't see them in equation 9. Change a form of this sentence, because definition of two constants is presented not in eq.9 but in line 194.
Other remarks (minor corrections):
Equation 34: Some of the derivative dots are in the wrong place. Please take a look and correct it if necessary.
Manuscript needs some corrections, see line 7 for example - additional commas etc.
Please check the name of your university. Is it written in lowercase?
Before publication, the article must be corrected to increase its readability.
Reviewer 3 Report
Comments and Suggestions for Authors
1) Provide the source of equations if they are from other sources. For instance, Equation (1). Or else, the authors need to elaborate on how to derive the equations. The same comments applies to lemmas and theorems that are mentioned without proof; for instance, lemma 1 and lemma 2.
2) Stability analysis in Subsection 3.5 is based on quadratic Lyapunov functions. As shown in [R1] and [R2], using non-quadratic Lyapunov function in adaptive schemes leads to better results in terms of tracking performance. The authors need to discuss this very important point as a remark in the manuscript.
[R1]. "Performance enhanced model reference adaptive control through switching non-quadratic Lyapunov functions", 2015. [https://doi.org/10.1016/j.sysconle.2014.12.001]
[R2]. "Model reference adaptive control with L^{1+α} tracking", 1996. [https://doi.org/10.1080/00207179608921661]
3) Zeno behavior needs to be discussed first in Subsection 3.6, before conducting the feasibility analysis.
4) This manuscript lacks a proper comparison study with prior work. Consider comparing your method with a state-of-the-art algorithm; this would help the readers to understand pros and cons of the proposed method.
5) More details on the utilized RFBNN are required. How can one ensure that the combination of the RFBNN with the proposed method does not hamper stability and feasibility analyses?
Comments on the Quality of English LanguageAcceptable.
Round 2
Reviewer 3 Report
Comments and Suggestions for Authors
No further comment.
Comments on the Quality of English LanguageAcceptable.